# Concise gene signature for point-of-care classification of tuberculosis

Jeroen Maertzdorf[1],[**],[†], Gayle McEwen[1],[†], January Weiner 3rd[1], Song Tian[2], Eric Lader[2], Ulrich Schriek[3], Harriet Mayanja-Kizza[4], Martin Ota[5],[†], John Kenneth[6] & Stefan HE Kaufmann[1],[*]

## Abstract

There is an urgent need for new tools to combat the ongoing tuberculosis (TB) pandemic. Gene expression profiles based on blood signatures have proved useful in identifying genes that enable classification of TB patients, but have thus far been complex. Using real-time PCR analysis, we evaluated the expression profiles from a large panel of genes in TB patients and healthy individuals in an Indian cohort. Classification models were built and validated for their capacity to discriminate samples from TB patients and controls within this cohort and on external independent gene expression datasets. A combination of only four genes distinguished TB patients from healthy individuals in both cross-validations and on separate validation datasets with very high accuracy. An external validation on two distinct cohorts using a real-time PCR setting confirmed the predictive power of this 4-gene tool reaching sensitivity scores of 88% with a specificity of around 75%. Moreover, this gene signature demonstrated good classification power in HIV[+] populations and also between TB and several other pulmonary diseases. Here we present proof of concept that our 4-gene signature and the top classifier genes from our models provide excellent candidates for the development of molecular point-of-care TB diagnosis in endemic areas.

**Keywords** disease classification; genomics; molecular diagnosis; real-time PCR; tuberculosis
**Subject Categories** Biomarkers & Diagnostic Imaging; Chromatin, Epigenetics, Genomics & Functional Genomics; Microbiology, Virology & Host Pathogen Interaction

## Introduction

With an estimated 9.6 million new cases annually, tuberculosis (TB) remains one of the major threats to global health (World Health Organization, 2015). A major obstacle to both controlling the spread of the disease and improving treatment outcome is the lack of efficient clinical tools for rapid and accurate diagnosis. New diagnostics could reduce TB incidence rates by 13–42%, with nucleic acid amplification tests preventing 23% of TB-related deaths in the Southeast Asian region (Abu-Raddad et al, 2009). Despite the broad extent of clinical experience with TB disease, a substantial proportion of new cases are mis- or undiagnosed, with diagnosis largely relying on detection of the *Mycobacterium tuberculosis* (Mtb) pathogen in sputum using smear microscopy, culture, or molecular techniques (Pai & Schito, 2015). In cases where sputum smear results are negative or when sputum samples cannot be obtained, cases of active TB remain recalcitrant to diagnosis (Norbis et al, 2013).

In order to explore the molecular mechanisms of TB infection and to discover potential diagnostic biomarkers, global gene expression profiles in peripheral blood have been investigated (Maertzdorf et al, 2015). Over the past decade, a substantial number of RNA transcriptional profiling studies harnessing microarray-based technologies have been published, identifying clusters of genes that are differentially expressed between TB patients and healthy individuals (Weiner et al, 2013). Most of these studies show highly consistent gene expression patterns that can be used to classify TB patients from healthy controls. For example, Berry et al (2010) described a whole-blood 393-transcript signature for detection of active TB and a specific 86-transcript signature that can discriminate active TB from other inflammatory and infectious diseases. More recently, Kaforou et al (2013) detected a 44-transcript signature that can distinguish TB from other diseases and a 27-transcript signature that distinguishes TB from latent TB infection (LTBI). The latter study also included TB patients from both HIV[−] and HIV[+] populations. Bloom et al (2013) described a 144-transcript signature that could distinguish TB from other pulmonary diseases, including pneumonia and lung cancer.

1 Max Planck Institute for Infection Biology, Berlin, Germany
2 Qiagen, Frederick, MD, USA
3 Qiagen GmbH, Hilden, Germany
4 Makerere University, Kampala, Uganda
5 Medical Research Council, Banjul, The Gambia
6 St. John's Research Institute, Bangalore, Karnataka, India
*Corresponding author. Tel: +49 30 28460502, E-mail: kaufmann@mpiib-berlin.mpg.de
**Corresponding author. Tel: +49 30 28460514, E-mail: maertzdorf@mpiib-berlin.mpg.de
†These authors contributed equally to this work

 

However, with signatures comprising dozens or hundreds of genes, and the prohibitive cost and expertise required to use microarray technology in the clinic, particularly in resource-poor settings, no diagnostic tools for TB based on gene expression have so far been developed. In contrast to microarray technologies, modern real-time reverse transcriptase PCR (RT–PCR)-based tools have several advantages for clinical use, being fast, cheap, easy to use, and requiring minimal electric power. Although molecular assays have their drawbacks for routine diagnosis as compared to more basic laboratory tests, RT–PCR assays are widely approved for detection of influenza viruses (http://www.cdc.gov/flu/professionals/diagnosis/molecular-assays.htm) and have been extensively developed over recent years for diagnosis of several diseases caused by bacteria (Maurin, 2012). Moreover, modern microfluidics technologies have enabled the design of "lab-on-a-chip" systems for RT–PCR detection of viral infections (Lee *et al*, 2008; Song *et al*, 2012). Additionally, because of its simplicity and rapidity, another amplification technique known as loop-mediated isothermal amplification (LAMP; Notomi *et al*, 2000) is also applied to detection of, for example, pathogenic microorganisms (Fu *et al*, 2011). We envisage that with such technologies, a new tool for diagnostic triage of TB patients is also in reach.

In order to develop such a tool, a small signature of genes that can robustly discriminate TB patients from healthy individuals is required. Here we describe the identification of such a small size gene signature that can discriminate TB patients from healthy individuals with high sensitivity and specificity. In this study, we designed a targeted RT–PCR array based on two microarray datasets from a South African and a Gambian TB cohort previously generated by our group (Maertzdorf *et al*, 2011a,b) and then applied it to a new Indian cohort to generate expression data for training and testing validation of TB classifier models. We used two tree-based methods to generate models for TB classification, namely random forest (RF) and conditional inference (CI) trees. We applied a stepwise approach to define a small top set of classifier genes that finally resulted in a minimally sized signature of only four genes that could successfully discriminate TB patients from healthy controls. For validation, we carried out RT–PCR of this signature and measured its classification performance on two separate African cohorts. Furthermore, using independent publicly available blood transcription microarray datasets (Berry *et al*, 2010; Maertzdorf *et al*, 2012; Bloom *et al*, 2013; Kaforou *et al*, 2013; Dawany *et al*, 2014), we demonstrate that our signature is capable of distinguishing between TB and other pulmonary and inflammatory diseases, across platforms and genetic backgrounds. Our use of RT–PCR readout in this new dataset allowed us to show that specific primer pairs for these genes have adequate dynamic range and so are appropriate for use in a diagnostic setting. These genes provide proof of principle for the development of a simple diagnostic point-of-care test in TB.

## Results

### RT–PCR array and cohort design

In this study, we aimed to identify a minimal set of genes that could be implemented into a clinical tool for point-of-care testing for TB. Since nucleic acid amplification techniques are readily applied in standard diagnostic settings, we evaluated the discriminatory power of a selected set of genes to distinguish between TB patients and healthy individuals using a targeted RT–PCR validation strategy. Based on previously published microarray datasets from South Africa and The Gambia (Maertzdorf *et al*, 2011a,b), we selected a set of 360 target genes that show strong differential expression between TB patients and healthy controls. From a cohort in Bangalore, India, a total of 200 peripheral blood RNA samples were collected from 120 TB patients and 80 healthy donors (60 LTBI and 20 uninfected). This new set of samples served as the primary source of data to evaluate the selected genes and to validate small gene sets for classification.

Gene expression levels for the selected target genes plus 12 reference genes were analyzed using a custom RT–PCR array. To maintain a balanced design with a 3:2 ratio of TB patients to healthy controls, 60 patient and 40 control samples were randomly assigned into a training set, with the remaining samples being retained as validation set.

Principal component analysis (PCA) was applied to the normalized gene expression levels to inspect the variation in the dataset. The results indicate that the gene expression differences between TB patients and healthy controls constitute the majority of the variance within the dataset as indicated by their separation in the first two principal components (Appendix Fig S1). For training and testing of machine learning models for disease-state classification, we randomly assigned 60 patient and 40 control samples to a training set, maintaining the ratio of 3:2 of the whole dataset. The remaining samples were retained as a test set.

The overall sample assignment and analysis steps applied in this study were as follows: (i) training set used to generate models and test them on the test set, (ii) determine optimal biosignature sizes and model training approaches, (iii) merge test and training set to maximize model performance, (iv) test full models using internal cross-validation, (v) test full models on qPCR data from external cohorts, and (vi) test full models on independent (microarray) data (Fig 1).

### Model training and validation on RT–PCR dataset

To initially investigate the number of genes that are required for good discrimination of TB patients from healthy controls in our dataset, RF models were built multiple times on the training set (Fig 1) and each time the genes were ordered by their relative importance as predictors (Gini importance index). Within each iteration, models based on varying numbers of the top genes (from 1 to 100 genes) were tested for their performance in a 25-fold cross-validation on the training set only. The resulting classification performance of models based on increasing numbers of genes was discerned using the area under the curve (AUC) statistic. We found that the number of genes within a model did not have a dramatic effect upon the classification performance over a large range of genes tested (Fig 2), suggesting that only a limited set of top-ranking genes already suffices for a high classification performance while additional genes do not significantly add more power to the performance of the models.

To identify the top classifier genes within our dataset we used RF models to rank genes according to their importance in partitioning the samples into the defined groups (TB vs. healthy). Generating 25

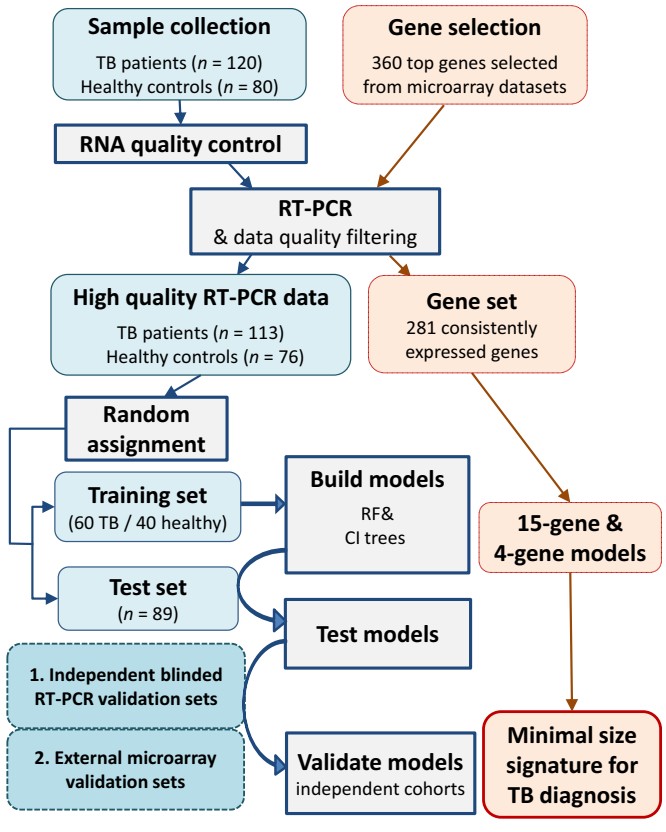

**Figure 1.  Sample and analysis flowchart.**
All samples from TB patients and healthy controls were quality controlled and randomly assigned into a training and test set. Built classification models were tested in both the test set and the total dataset. External validation was performed on two independent blinded RT–PCR datasets, as well as on external microarray datasets.

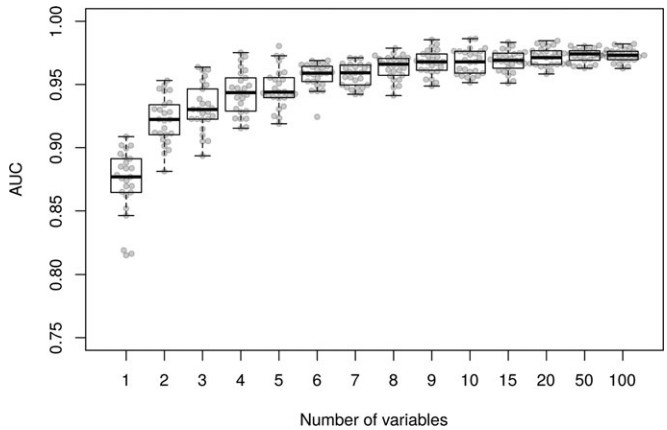

**Figure 2.  Model size and performance.**
Classification performance of varying numbers of top-ranking genes in RF models built and cross-validated on the training set. Displayed are AUC values from 25 reiterative RF models with boxplot overlay.

returned comparable high performance results, with AUCs 0.98 (CI = 0.96–1.0) and 0.97 (CI = 0.94–1.0), respectively (Fig 2).

Whereas random forests can output a list of genes ordered by their importance as classifiers, CI trees output a set of top predictor genes, a single simple decision tree and also have the advantage of generating easily interpretable predictor values. Intriguingly, running such a CI decision tree on samples in the training set resulted in a very small set of only three genes. Calculation of the classification power of these genes on the test set RT–PCR data gave a performance of AUC = 0.95 (95% CI: 0.90–0.99).

**Models based on full RT–PCR dataset**

In the following step, we merged the training and tests sets from the Indian cohort to generate models to be tested both using internal cross-validation, and by applying them to external datasets.

Since large training sets usually provide better classification power, we also evaluated a 15-gene signature built on the whole 189 sample dataset (both training and test sets; Appendix Table S1). The new model arising from this whole-dataset RF classification analysis contained largely the same genes as the one based on the original 100-sample training set. Ranked genes within this 15-gene classification signature are given in Appendix Table S2. The excellent classification performance of this new model in an internal cross-validation procedure (AUC = 0.98, accuracy (ACC) = 0.94) was very similar to the predictive power of the original 100-sample training set model (AUC = 0.98, ACC = 0.91). There was no difference in the classification power of this signature when separating controls into LTBI and uninfected individuals (Fig EV2).

We also evaluated a CI decision tree using the whole dataset to achieve maximum classification power within our RT–PCR dataset. This approach resulted in a very small signature, comprising only four genes. This 4-gene model reached very high classification performance scores with AUC = 0.98 and ACC = 0.92. Classification power was the same for predicting TB from LTBI or uninfected healthy controls (AUC = 0.98 for both (Fig EV2)). A graphical display of the CI decision tree (Fig 4) also illustrates the advantage

replicates of an RF model on the whole dataset we counted how many times each gene was among the most important predictors. Twelve genes ranked within the top 15 in all RF models (Fig 3A). Variable importance (mean decrease in Gini coefficient) for each gene was used as a measure for its predictive importance in partitioning the samples into the defined groups (Fig 3B). In this context it thus means that a gene with a higher variable importance plays a greater role in predicting whether a sample is from a TB patient or a healthy individual. Below the top 12 genes, the importance measure shows a break-off point. The relative expression distributions of these top classifier genes showed mostly higher expression levels in TB patients (Fig 3C). Expression levels in LTBI and uninfected controls were highly similar (Fig EV1).

Based on the above results (Fig 2), one can see that the 15-gene signatures are well within range of the maximum predictive performance in our training dataset. It is also interesting that a smaller number of genes can also achieve a high level of predictive power. However, in order to not lose sensitivity by including too few genes at this stage of the performance testing, we chose to initially use a 15-gene RF model to run on the test set (Fig 1). The classification performance of this model on the test set returned an AUC as high as 0.98 (CI = 0.97–1.0), documenting the model's high overall performance. Testing two other models based on 10 and 5 genes

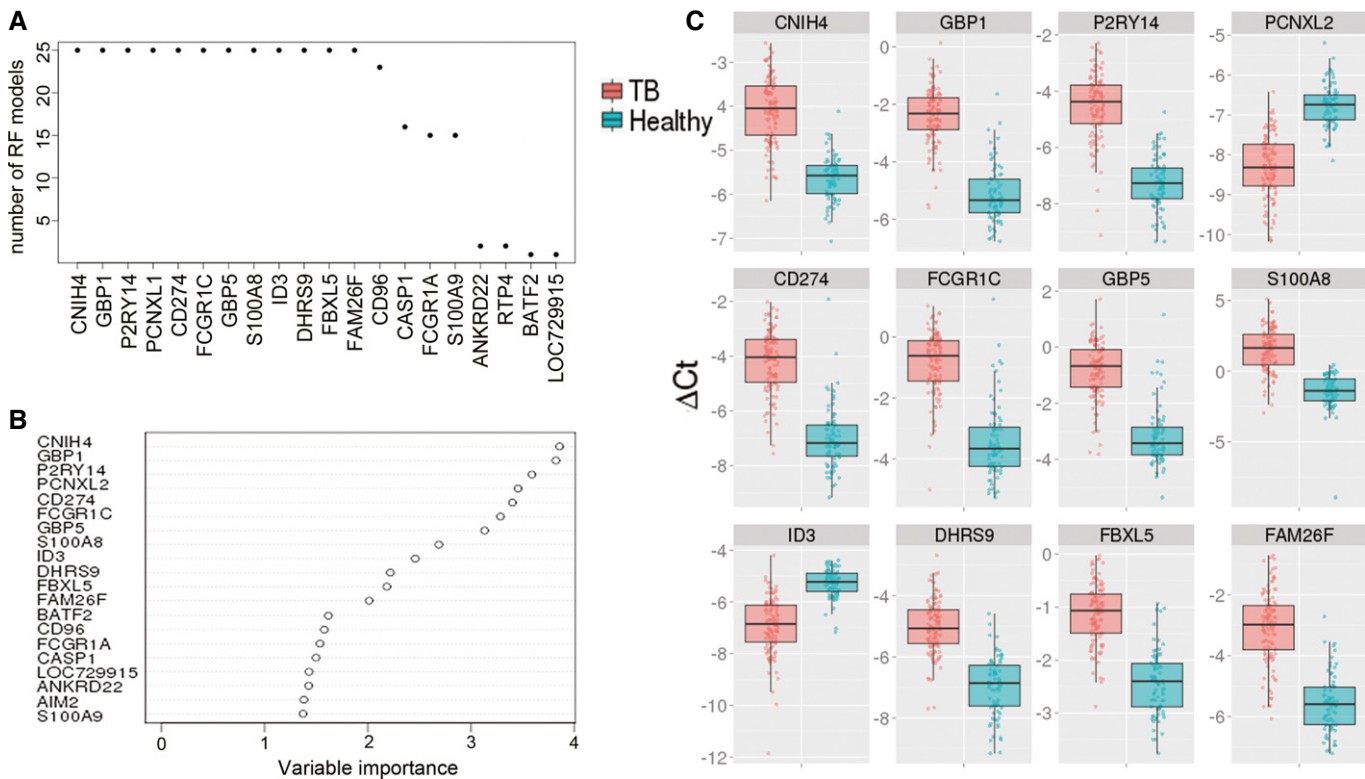

**Figure 3. Top classifier genes.**

A   Cumulative number of times that genes were present among the top 15 ranking genes from 25 reiterative RF models.
B   Variable Importance (Gini) for top 20 genes from RF model that was used to define the 15-gene signature.
C   Box plots showing relative expression levels of top classifier genes in our RT–PCR dataset. Displayed are inverse deltaC*t* values (zero minus ΔC*t*), such that higher values indicate higher expression levels for each sample in this study.

of easily interpretable predictor values generated by this CI decision tree. For example, samples showing a ΔC*t* value lower or equal to −4.1 for GBP1 and higher than 5.9 for ID3 are most likely from healthy individuals, whereas those showing > −4.1 for GBP1, > 6.7 for P2RY14 and > −3.5 for IFITM3 almost certainly originate from TB patients. The excellent classification performance of this 4-gene model indicates that very small gene signatures can have very high predictive power. However, to determine if these gene models are of broad application, we validated these gene models on independent datasets as well.

**Blind RT–PCR validation on external cohorts**

To validate our signature in a RT–PCR-based experimental setting, we tested the performance of our 4-gene model on whole-blood RNA samples collected from two external cohorts. RT–PCR of the 4-gene signature was carried out on a total of 75 samples from The Gambia and 62 samples from Uganda (see Appendix Table S1 for number of TB, LTBI, and uninfected donors). Using the 4-gene model, trained and tested on the Indian cohort, the TB probability scores (ranging from 1 to 0) were used to predefine two cutoff points (0.8 and 0.6), aimed at reaching higher specificity or higher sensitivity, respectively. The 4-gene model was then applied in a blinded manner to predict TB probability scores for each sample.

We found that the model performed well on the two external cohorts, with the cutoff of 0.8 giving sensitivity and specificity scores of 85% and 76%, respectively, in the Gambia cohort (AUC = 0.89), but a somewhat lower sensitivity (73%) but equal specificity (78%) in the Uganda cohort (AUC = 0.82). The cutoff of 0.6 on the other hand performed better for the Uganda cohort (87% sensitivity, 75% specificity), while for the Gambian cohort sensitivity slightly increased (88%), although with lower specificity (68%).

From the receiver operator characteristic (ROC) curves for the 4-gene model performance in the Ugandan and Gambian cohort (Fig EV3) one can see that the predictions are very similar for both cohorts. Any point along these curves represents different combinations of sensitivity and specificity scores for each cohort, based on the cutoff for TB probability threshold scores set. No difference was observed in the classification performance of TB vs. LTBI or uninfected controls (Appendix Fig S2). The performance in the TSTneg group even seemed a bit weaker than in the TSTpos group in both cohorts, but this is mostly due to the lower number of uninfected compared to LTBI individuals in both datasets.

**Validation on other independent datasets**

A robust gene signature that can be efficiently used as a diagnostic triage tool at point of care should be able to accurately discriminate

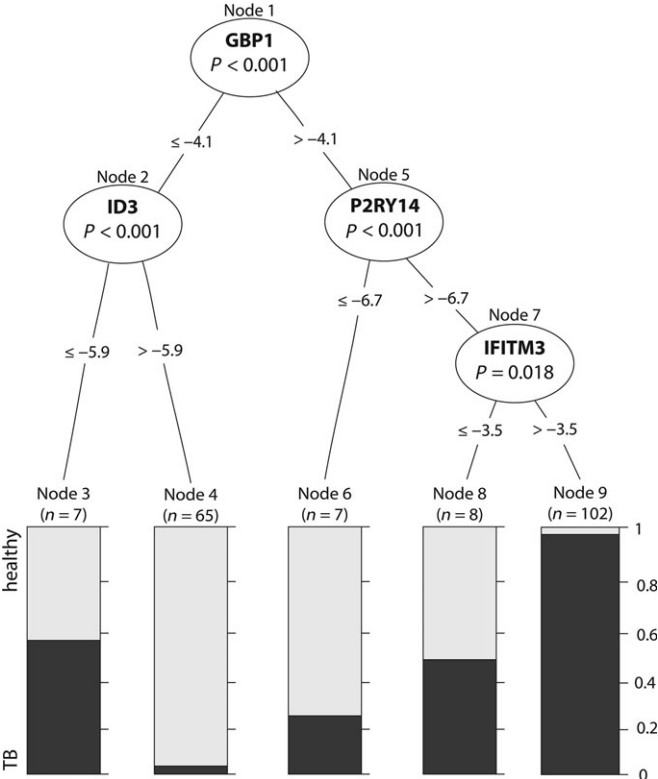

**Figure 4. Conditional inference tree built on the whole dataset.**
Decision tree for the 4-gene signature was built using ctree function from the party package in R. Significant predictor genes are displayed in oval nodes. Numbers between nodes indicate inverse $\Delta Ct$ values for each split. Terminal nodes display the relative proportion of samples from TB (dark gray) and healthy controls (light gray).

TB patients from healthy individuals across a wide range of ethnic backgrounds and also across different technology platforms. Ideally, it should also not be confounded by other diseases or environmental factors. To this end, we tested our gene models for their classification power in several independent, publicly available microarray expression datasets (Berry et al, 2010; Maertzdorf et al, 2012; Bloom et al, 2013; Kaforou et al, 2013; Dawany et al, 2014). These publicly available datasets include expression profiles from TB patients and healthy controls, as well as from patients with other diseases (ODs) than TB in both HIV$^-$ and HIV$^+$ populations.

Table 1 summarizes the overall classification performance of the 4-gene and 15-gene models based on all RT–PCR and the independent microarray datasets.

In each independent validation dataset we were able to robustly classify individuals with TB from healthy controls in HIV$^-$ individuals using both models; in all cases an AUC of between 0.91 and 0.99 was observed (Fig 5). Again, no significant differences in classification performance were observed between LTBI and uninfected controls (Appendix Fig S3). Even predicting TB in HIV-co-infected populations resulted in robust classification performance, e.g. AUC of 0.84 for the 4-gene model and 0.89 for the 15-gene set in the Kaforou dataset (Kaforou et al, 2013) (Fig 5).

When we included ODs as confounding factors in our predictions, our models still performed well (Table 1), despite being

trained on data from TB patients and healthy controls only. Indeed, in distinguishing TB from ODs, the 4-gene set performed well and surprisingly even outperformed the 15-gene set on several other pulmonary diseases (Fig EV4). This is probably due to the fact that using a larger gene set results in the inclusion of more genes that are related to general inflammation, creating a larger overlap with expression profiles in non-TB inflammatory ODs. The Bloom dataset (Bloom et al, 2013) contains individuals with other pulmonary diseases (sarcoidosis (SARC), lung cancer and pneumonia) and the Maertzdorf dataset (Maertzdorf et al, 2012) includes TB and SARC patients. The 4-gene model performed very well in distinguishing TB from pneumonia and lung cancer, but performed less well in distinguishing TB from SARC in the Bloom dataset (AUC = 0.72) (Fig EV4). In contrast, TB could not be distinguished from SARC in the Maertzdorf dataset although this could be due in part to the small number of samples. Alternatively, the Bloom dataset also included non-active SARC patients, while the Maertzdorf dataset only contained SARC patients with active disease status. The Kaforou dataset included multiple ODs where TB was initially considered in the differential diagnosis; here our 4-gene model performed reasonably well, even in an HIV-co-infected population (both AUC = 0.71). In distinguishing TB from non-pulmonary diseases in the Berry dataset (Berry et al, 2010) the 4-gene model was also seen to perform well (Fig EV4).

## Discussion

The continuing TB pandemic results in 1.5 million deaths every year, and is particularly devastating in resource-poor countries and countries where HIV is also highly prevalent (World Health Organization, 2015). The ability to diagnose TB rapidly is imperative for combating this deadly disease as prompt diagnosis improves treatment outcomes and also prevents prolonged spread. An estimated 13–42% reduction in TB incidence cases could be achieved by implementing new diagnostic methods (Abu-Raddad et al, 2009).

We have identified a number of top classifier genes and a minimally sized signature with high classification power, which provides excellent gene candidates for inclusion in a diagnostic point-of-care test in endemic regions. Our signatures are capable of distinguishing TB patients from healthy individuals with very high accuracy on a RT–PCR validation set. We consider this validation on RT–PCR a key part of our results, since it proves that small triage gene sets work in a simple RT–PCR setup. Such a platform could be integrated in a clinical setting, for example using a small gene-chip device, something that is not possible with complex microarray platforms.

For a true external validation, we tested the performance of our signature by an RT–PCR-based platform in two separate populations from The Gambia and Uganda. Depending on the chosen cutoff point, these validations reached sensitivity scores up to 87% and with a specificity around 75%. Due to technical differences between sites and varying gene expression levels between ethnic populations, optimal cutoff settings could thus vary between different locations. Calibration of equipment in different settings would therefore probably predefine different cutoff points for different cohorts or geographical regions. Moreover, defining the desired cutoff is mostly a clinical decision, whether the TB predictive power should

**Table 1.  Classification performances of signatures on RT–PCR datasets and external microarray datasets used in this study.**

| Platform | Classification | Cohort | N | 4 gene AUC (95% CI) | 15 gene AUC (95% CI) |
|---|---|---|---|---|---|
| RT–PCR | TB vs. healthy | India (test set) | 89 | 0.95 (0.90–0.99) | 0.98 (0.97–1.00) |
| | | India (full dataset) | 189 | 0.98 (0.97–1.00) | 0.98 (0.96–1.00) |
| | | The Gambia | 75 | 0.89 (0.81–0.96) | *not done* |
| | | Uganda | 62 | 0.82 (0.71–0.93) | *not done* |
| Microarray | TB vs. healthy (HIV⁻) | Bloom | 148 | 0.99 (0.98–0.99) | 0.98 (0.97–0.98) |
| | | Berry | 228 | 0.91 (0.86–0.91) | 0.92 (0.88–0.92) |
| | | Kaforou | 180 | 0.91 (0.87–0.91) | 0.96 (0.94–0.96) |
| | | Maertzdorf | 26 | 0.99 (0.95–0.99) | 0.99 (0.97–0.99) |
| | TB vs. healthy (HIV⁺) | Kaforou | 182 | 0.84 (0.79–0.84) | 0.89 (0.84–0.89) |
| | | Dawany | 44 | 0.72 (0.56–0.72) | 0.80 (0.66–0.80) |
| | TB vs. ODs (pulmonary) | Bloom (pneumonia) | 49 | 0.94 (0.87–0.94) | 0.68 (0.51–0.68) |
| | | Bloom (lung cancer) | 51 | 0.95 (0.88–0.95) | 0.78 (0.65–0.78) |
| | | Bloom (sarcoidosis) | 96 | 0.72 (0.62–0.72) | 0.71 (0.60–0.71) |
| | | Maertzdorf (sarcoidosis) | 26 | 0.58 (0.34–0.58) | 0.63 (0.40–0.62) |
| | | Kaforou (others) | 180 | 0.71 (0.63–0.71) | 0.63 (0.55–0.63) |
| | TB vs. ODs (non-pulmonary) | Berry (Still's disease) | 85 | 0.85 (0.77–0.85) | 0.78 (0.68–0.78) |
| | | Berry (ASLE) | 82 | 0.83 (0.75–0.83) | 0.87 (0.80–0.87) |
| | | Berry (PSLE) | 136 | 0.75 (0.66–0.75) | 0.71 (0.61–0.71) |
| | | Berry (staphylococcus) | 94 | 0.88 (0.81–0.88) | 0.80 (0.71–0.80) |
| | | Berry (streptococcus) | 66 | 0.93 (0.87–0.93) | 0.87 (0.79–0.87) |
| | TB vs. ODs (HIV⁺) | Kaforou (others) | 190 | 0.71 (0.63–0.71) | 0.63 (0.55–0.63) |

"Cohort" indicates name for each dataset. *N* is the number of subjects in each dataset. AUC, area under curve; CI, confidence interval; TB, tuberculosis; ODs, other diseases. Major ethnicities of the validation cohorts are as follows: Bloom: Afro-Caribbean, SE Asian, Hispanic, Middle Eastern, Indian, Black, Caucasian; Berry: Asian, Black, Caucasian; Kaforou: Black (South and East African); Maertzdorf: Caucasian; Dawany: Black (South African). Accession numbers for each dataset are given in the Material & Methods section.

yield high sensitivity (with lower specificity) or whether higher specificity would be preferred (at the cost of sensitivity).

Overall, the 4-gene model displayed the best performance, with similar AUCs to the 15-gene model for discriminating between TB patients and healthy individuals, but far better performance when other diseases were considered. This small set of four genes consisted of GBP1, IFITM3, P2RY14 and ID3. Both GBP1 and IFITM3 are interferon-induced genes, corresponding to the strongly upregulated interferon signatures observed in TB patients (Berry *et al*, 2010). However, these genes are not in the same gene modules as defined by Chaussabel & Baldwin (2014), indicating that transcriptional profiles of GBP1 and IFITM3 are probably not highly correlated. The other two genes are not known to be specifically involved in infectious diseases, although P2RY14, a G-protein-coupled receptor with diverse physiological roles, has been implicated in the modulation of immune function (Scrivens and Dickenson, 2005). ID3 is a T lymphocyte associated transcriptional regulator (Naito *et al*, 2011). Note that three genes within this 4-gene signature, with the exception of IFITM3, are among the top 12 classifier genes as defined in Fig 2 and were also present in our 15-gene model (IFITM3 ranked at position 47 of most important predictors in the RF model on which our 15-gene signature was based). We presume that this is due to the fact that several of the top-ranking genes have

correlated expression patterns. Therefore, the combination of genes that results in a good classification model does not necessarily need to correspond to the top-ranking predictor genes.

While this study was not set up with the aim to discriminate TB from other diseases, we nevertheless discovered that our 4-gene signature could distinguish such cases with very high AUCs for some diseases (Table 1 and Fig EV4), better or equal to the published transcriptome signatures. Within the Kaforou, Berry and Bloom datasets (Berry *et al*, 2010; Bloom *et al*, 2013; Kaforou *et al*, 2013) our 4-gene signature was also able to distinguish TB from other adult pulmonary diseases, as well as non-pulmonary bacterial diseases, systemic inflammatory diseases and including pediatric patients. Our 4-gene signature performed especially well in distinguishing TB from pneumonia and lung cancer. Our signatures failed, however, to distinguish between TB and sarcoidosis in our own, relatively small, dataset (Maertzdorf *et al*, 2012), but did show some classification power within the Bloom dataset (Bloom *et al*, 2013). This was not surprising, since these two pulmonary diseases are clinically, pathologically and immunologically highly similar despite their distinct etiologies. However, due to the relatively low prevalence of sarcoidosis in TB-endemic regions (Babu, 2013), this overlap will not markedly influence the use of our signatures in possible future diagnosis of TB.

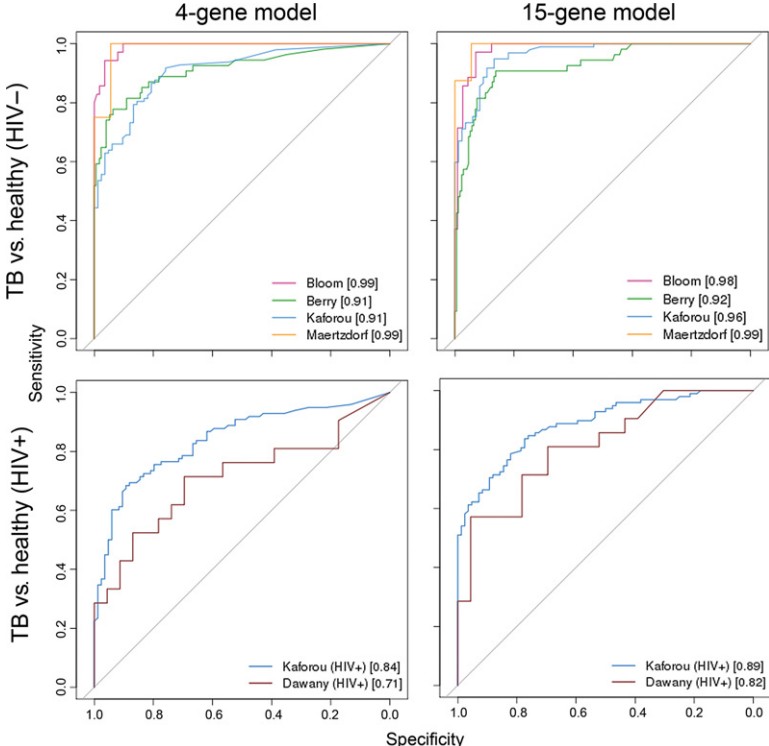

**Figure 5.    Classification performance on external datasets.**
ROC curves visualizing the classification performance of our 4-gene and 15-gene set models on external microarray datasets in both HIV⁻ and HIV-co-infected individuals. Left panels, 4-gene signature; right panels, 15-gene signature. Upper row shows classification performance between TB and healthy in HIV⁻ populations; lower row in HIV-co-infected populations. Numbers in parentheses are AUC values. For detailed classification measurements refer to Table 1.

Indeed, the top-ranking genes in our classification signatures appear to mainly reflect host pathology and are not necessarily specific for TB. However, we find that our small signature does discriminate some other pulmonary diseases with high accuracy, despite not being based on data from such diseases. Indeed, the 4-gene signature performed remarkably better than the extended 15-gene set, likely due to the fact that the larger gene set included more general inflammatory signatures, leading to a higher overlap in gene expression profiles with other inflammatory diseases.

We would like to re-emphasize that our study was initially set up to classify TB patients from healthy individuals, and not for differential diagnosis of suspected disease. In our opinion, a simple "one size fits all" signature that would be able to distinguish TB from healthy and, at the same time, differentially diagnose TB from other (pulmonary) diseases is unlikely to emerge. We rather envision an easy two parallel signature setup that when (i) the signature classifying healthy from probable TB tests positive, then (ii) a second differential diagnosis signature (yet to be defined) would distinguish TB from other diseases.

A similar effort to identify diagnostic gene expression signatures for TB has recently been published (Satproedprai *et al*, 2015). A smaller cohort of TB patients and healthy controls compared to the one presented here (78 samples compared to 200 samples), and a smaller set of genes (13 compared to 360 here) were evaluated by RT–PCR. In this study, gene expression levels were used to calculate a "TB sick score" to predict TB. The selected genes could distinguish

TB patients from healthy controls within their cohort, but the signature was not validated on an independent cohort. Moreover, while the selected genes were among the differentially expressed genes identified by Berry *et al* (2010), they were not within the 86 TB-specific transcript set. Accordingly, discrimination of TB from other diseases may not be possible with the signature of Satproedprai *et al* (2015), although this remains to be tested.

Overall, we demonstrate in an exploratory setup, proof of concept that small sets of genes like our 4-gene signature can offer excellent classification performance in an easily implemented RT–PCR setup. Therefore, these gene sets could be used for a potential future point-of-care screening of active TB patients. We believe that such a limited set of genes could soon be implemented in a tool for large-scale diagnostic validation in endemic regions to test its potential to diagnose active TB. Although such a new test will not immediately replace existing diagnostics like sputum smear culture and GeneXpert (Boehme *et al*, 2011), proof of principle is provided here that simple and rapid RT–PCR-based tests can help to classify TB patients.

## Materials and Methods

### Study design

A prespecified total of 200 adult participants (120 TB patients and 80 healthy donors) were recruited at St. John's hospital in Bangalore,

India. Inclusion criteria for patients were cases of newly diagnosed active TB, confirmed by a positive GeneXpert sputum test (Boehme *et al*, 2011) and treatment not yet initiated. Healthy control individuals showed no symptoms or signs of active TB and were age and gender matched to the recruited TB patients. Exclusion criteria were subjects under 18 years of age or over 60, HIV$^+$ status, and individuals on treatment for TB at the time of recruitment or having received such treatment within the last 12 months. Healthy control individuals were recruited among local healthcare workers and tested for LTBI by skin test and IGRA test (Quantiferon-TB). 29% of skin test-negative individuals were tested IGRA$^+$, while 15% of skin test-positives tested IGRA$^-$. LTBI individuals were defined based on positive skin test to match the definition for latency in the other validation cohorts. Numbers of patients, LTBI and uninfected individuals are given in Appendix Table S1.

Peripheral blood (2.5 ml) was drawn into PAXgene tubes and RNA was extracted using the PAXgene Blood miRNA extraction kit (Qiagen) following the manufacturer's instructions. Low quality RNA samples were excluded from the study (seven samples with A260/280 < 1.6). For this validation study, 360 target genes were selected based on our previously published microarray datasets from South Africa and The Gambia, using a ranking strategy based on the original analysis and RF variable importance (Maertzdorf *et al*, 2011a,b). Gene expression of these targets plus 12 reference genes was determined using a custom 384 well RT$^2$ Profiler RT–PCR array (Qiagen). Samples were randomly assigned to a training set of 60 TB cases and 40 controls, maintaining the study design's 3:2 ratio of TB patients to controls. The remaining samples were assigned to a separate test set. No predefined analysis protocol was used in this study.

## Data analysis

The entire raw dataset was evaluated for quality control. Genes with C*t* values > 32 in at least 40% samples in both TB and control groups, and more than 50% of the samples in either group were dismissed. These genes were considered to have an extremely low expression level and therefore not useful for diagnostic-level classification and excluded from the analysis. This resulted in a final set of 281 genes. The stability of the reference genes was evaluated with the Bioconductor geNorm analysis R package NormqPCR (Perkins *et al*, 2012). The geometric mean (GEOMEAN) of the C*t* values of the top four selected reference genes (RPLP0, EEF1A1, UBE2D2 and B2M) was calculated for each sample as its normalization factor.

## Statistics

Differential expression of genes was calculated using a simple *t*-test in R, with correction for multiple testing controlling the false discovery rate (Benjamini & Hochberg, 1995).

RF models were generated using the randomForest package in R (Liaw & Wiener, 2002) and CI trees using the *ctree* function from the *party* package in R (Hothorn *et al*, 2006). We ran both these methods on the training sets and then built models based on the top genes to run on the test set. We also ran both these methods on our entire dataset (training and validation sets combined) as

### The paper explained

**Problem**

Tuberculosis (TB) continues to have a major impact on global health, particularly in resource-poor countries. There is a great need for efficient clinical tools for rapid diagnosis in order to better control the ongoing disease burden. Multiple studies have shown that gene expression signatures can be used to discriminate TB patients from healthy individuals but, to date, these signatures have been large and not applicable for simple diagnostic application.

**Results**

We show that a gene signature containing as few as four genes can be used to rapidly diagnose probable TB cases. By using a standard RT–PCR assay, this signature could be directly applicable as a simple triage tool. Using cohorts from multiple geographical regions and ethnicities, we show that our signature can predict TB cases with high sensitivity and specificity, and even to some extent differentially diagnose them from several other pulmonary diseases.

**Impact**

The classification signatures we describe in this work could provide a valuable simple and rapid new diagnostic tool to help clinicians decide to exclude or include probable TB in suspected new cases. Such new diagnostics could potentially have a great impact on reducing TB disease incidence in high endemic countries.

larger datasets can provide more accurate predictions. All models were subsequently cross-validated on the full RT–PCR dataset and independently validated on RT–PCR data from two external cohorts (see Appendix Table S1 for number of patients and controls) and on five independent, publicly available gene expression (microarray) datasets. The performance of our models was visualized on ROC curves, evaluated through AUC statistics and their associated 95% confidence intervals using the pROC R package (Robin *et al*, 2011). Variable importance was measured by the mean decrease in Gini coefficient, calculated by the randomForest function.

For the validation of our gene signatures on independent microarray datasets, the datasets were first normalized as follows: We selected the healthy controls from each of the RT–PCR and microarray datasets. These controls were used to obtain medians and interquartile ranges for each gene in the set, and all samples from each dataset were standardized with these values. Next, RF models were trained on the normalized RT–PCR data and applied to the normalized microarray samples. Microarray datasets were downloaded from the Gene Expression Omnibus (GEO) database repository.

## Datasets

The following datasets, available in the Gene Expression Omnibus (GEO) database, were used in this study:

- RT–PCR data from the Indian cohort generated in this study: GSE74092
- External microarray validations sets: GSE42834 (Bloom *et al*, 2013); GSE19491 (Berry *et al*, 2010); GSE37250 (Kaforou *et al*, 2013); GSE34608 (Maertzdorf *et al*, 2012); GSE50834 (Dawany *et al*, 2014)

## Study approval

Ethical approval for gene expression analysis in peripheral blood in this study was issued by St. John's Medical College and Hospital Institutional Ethical Review Board, Bangalore, India (Ref No. 173/2012) and by the Ethical Board of the Charité Berlin (No. EA1/200/08). Informed consent was obtained from all study participants prior to inclusion in the study and experiments conform to the Declaration of Helsinki principles.

**Expanded View** for this article is available online.

## Acknowledgements

We thank all study participants and clinical and technical staff for providing and processing the samples. We would also like to acknowledge all other collaborators from Qiagen's bioinformatics team in Frederick, MD, for analyzing the data and M.L. Grossman for copyediting this manuscript. This work was supported by Qiagen and by the Norwegian Agency for Development Cooperation (NORAD).

## Author contribution

This study was designed by JM, ST, EL, US, JK, and SHEK. JM and GME performed most of the research. Clinical samples and data were provided by HMK, MO, and JK. Data analysis was performed by JM, GME, JW, ST, and EL. The manuscript was written by JM, ST, JW, GME, EL, and SHEK. All authors approved the manuscript.

## Conflict of interest

The authors declare that they have no conflict of interest.

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
