## [Review Process File · EMBO Molecular Medicine]

Concise gene signature for point-of-care classification of tuberculosis

Jeroen Maertzdorf, Gayle McEwen, January Weiner, Song Tian, Eric Lader, Ulrich Schriek, Harriet Mayanja-Kizza, Martin Ota, John Kenneth, Stefan HE Kaufmann

Corresponding author: Stefan Kaufmann, Max Planck Institute for Infection Biology

Review timeline:

Submission date:	01 July 2015
Editorial Decision:	06 July 2015
Resubmission date:	28 August 2015
Editorial Decision:	24 September 2015
Revision received:	19 October 2015
Editorial Decision:	11 November 2015
Revision received:	13 November 2015
Accepted:	17 November 2015

Transaction Report:

Editor: Céline Carret

1st Editorial Decision

06 July 2015

Thank you for the submission of your manuscript "Concise gene signature for point-of-care classification of tuberculosis".

I have now had the opportunity to carefully read your paper and the related literature and I have also discussed it with my colleagues. I am afraid that we concluded that the manuscript is not well suited for publication in EMBO Molecular Medicine and have therefore decided not to proceed with peer review.

We appreciate that your study proposes a novel molecular diagnosis tool to detect patients infected with TB from non infected using a simple PCR technique based on a minimum set of discriminatory genes. We do understand that such a tool would be much more helpful in the field than a gene-expression signature obtained from microarray or NGS analyses. While the data is convincing and well performed, we unfortunately regret that no sensitivity nor efficacy tests were done and compared to standard test(s) on a different cohort (even if smaller). Of relevance it would be very interesting to see how this technique performs in terms of specificity, sensitivity, time (from getting samples to diagnostic), etc.

Unfortunately, without this, the study remains at the discovery level and therefore does not quite fulfil our criteria for translational implications. Should you be willing and able to do more, I would however, be happy to reconsider my decision.

As it stands though, I am sorry that I cannot be more positive this time.

We hereby would like to resubmit our manuscript entitled “**Concise gene signature for point-of-care classification of tuberculosis**” by Jeroen Maertzdorf *et. al.*, to *EMBO Molecular Medicine*.

Based on your recommendations following our previous submission (manuscript EMM-2015-05598) and comments from other reviewers, we have performed additional validation work in two different cohorts to test the sensitivity and specificity performance of our RT-PCR –based tool in separate populations.

Tuberculosis (TB) remains one of the major threats to global health and new tools are urgently needed to combat the ongoing pandemic. Simple nucleic acid amplification-based platforms that can rapidly classify TB patients could be such a tool to help drastically reduce TB incidences and deaths. TB-associated gene expression profiles have been identified over the last years, but mainly using large gene sets and platforms that are only available in specialized laboratories.

We set out to validate the expression of key genes in a large new cohort from an TB-endemic region, applying a simple RT-PCR–based platform, which ensures the possibility of easy implementation in a clinical setting.

Here we present convincing evidence that a minimal set of only four genes gives excellent classification power to distinguish TB patients from healthy individuals. Using both separate validation sample sets and independent external datasets, we show the impressive predictive power of this small gene set to separate TB patients from healthy individuals across a wide range of ethnic backgrounds. Moreover, it can even distinguish TB from several other pulmonary diseases with high accuracy, thus minimizing confounding diagnoses.

Our findings provide proof of principle that a simple and rapid nucleic acid amplification test on a very limited number of target genes could work as triage tool to classify TB patients. The time is ripe for implementation and diagnostic validation of a small set of genes, such as the one presented here, in a simple clinical tool that can facilitate diagnosis of active TB patients at point of care in high-endemic regions.

We hereby confirm that this manuscript comprises original, unpublished material and has not been submitted elsewhere for publication. All authors have seen and approved the manuscript’s content, have reported no conflict of interest and have made significant contributions to the work described.

Thank you for the submission of your manuscript to EMBO Molecular Medicine. We have now heard back from the three referees whom we asked to evaluate your manuscript. Although the referees find the study of significant interest, they also raise a number of concerns that should be addressed in the next and final version of your article.

As you will see from the comments below, all three referees liked the study. However, while referee 1 is supportive and only requests additional details (a concern shared by referee 2), the other referees would like to see more testing of the robustness of the signature in a larger cohort with mixed ethnicity and pulmonary diseases status, including patients with LTBI. Finally, maybe a more random/blinded testing of a cohort would be of great interest and would clearly increase the translational potentials of the findings.

Given the balance of these evaluations, we will be happy to consider a revision of your manuscript and would like to strongly encourage you to address the robustness of the signature in additional patients.

Please note that it is EMBO Molecular Medicine policy to allow only a single round of revision and that, as acceptance or rejection of the manuscript will depend on another round of review, your responses should be as complete as possible.

I look forward to seeing a revised form of your manuscript as soon as possible.

***** Reviewer's comments *****

Referee #1 (Remarks):

This manuscript by Maertzdorf and co-workers describes the selection and use of human gene expression profiles for distinguishing tuberculosis (TB) patients from healthy individuals or patients with other pulmonary diseases. Based on data from large transcription profiling studies, the authors selected 4 genes, GBP1, IFITM3, P2RY14 and ID3 and obtained sensitivity scores of 88% with a specificity of around 75% to differentiate TB cases from healthy individuals. The selection of genes was based on signatures obtained from TB patients in Africa. Their profiles were then used for a training set and a validation set from patients from another geographic region (India). The authors also tested a 15 gene set and found that their selected four gene set provided equal or better discrimination of TB patients relative to patients with other diseases.

Overall, the study convincingly shows that RT-PCR results obtained from RNA preparations from patient's blood can help to reliably identify undiagnosed TB cases, suggesting that the described method could be developed into a promising diagnostic tool that might be used at the point of care units in addition to other tools for TB-case identification.

There was only one point that was not clear to me, which I may have missed: It was not clear to me which phenotypic criteria were originally used for defining individuals as TB patients or healthy individuals? Were the TB patients culture positive? Were the individuals tested by IGRA before being classified into the different categories of the study groups.

Referee #2 (Remarks):

Maertzdorf and colleagues extend the previous work conducted by them and others in identifying RNA signatures for active TB from whole blood. In this study they use a cohort of 120 active and 80 healthy controls from St. Johns Hospital in Bangalore, India. After curation they had 133 active TB and 76 healthy control samples which they divided into a training set (60 TB / 40 healthy) and validation set (53 TB / 36 healthy). They used both Random Forest and Conditional Inference tools to identify human transcripts with high predictive power. This led them to a 4 gene set and a 15 gene set. They further tested the 4 gene and 15 gene sets on 5 publically available studies: Berry (417 patients), Bloom (236 patients), Kaforou (327 patients), Maertzdorf (46 patients), and Dawany (36

patients).

The results show that the 4 gene set achieves ROC AUC values of 0.90-0.99 in HIV-negative patients with TB vs healthy patients. For HIV positive patients with TB vs healthy patients it was 0.72-0.81, and for either HIV-positive or negative patients with TB versus those with other diseases it was 0.63 - 0.84.

Comments:

1. The WHO has identified a strong need for a rapid POC triage test to identify patients at high risk for TB who should undergo further evaluation, so this work is of very high significance. However, in the field one would anticipate that patients presenting will be ill with a mixture of TB and other diseases. Hence the fact that the 4 gene predictor set was derived using TB patients versus healthy controls and the test's relatively poor performance in TB versus other diseases (AUC 0.63-0.84) is a major limitation of the study
2. The authors used the training subpopulation (60 TB / 40 healthy patients) to derive a 15 gene set. They then tested and further refined the 15 gene set with the whole cohort of both the training and the validation sets (133 TB / 76 healthy) in order to obtain the 4 gene set. They never tell us how the 15 gene test performs only on the validation set, and of course the validation set was partially utilized to identify the 4 gene set.
3. Please provide more information about the 80 healthy controls from Bangalore. Were they TST or IGRA positive for latent TB? How were they recruited? Were they actually patients seeking medical care or were they local students / health care workers?
4. Human genetics is likely to have a strong influence on human RNA-based diagnostics for TB. It would be helpful if the authors stated the ethnicity of the patients in the 5 publically available data sets used for validation in Table 1
5. Similarly, the degree of HIV immunosuppression is also likely to have a strong influence on human RNA-based diagnostics for TB. It would be helpful if the authors stated the CD4 counts of the HIV-positive patients in the 5 publically available data sets used for validation in Table 1.

Referee #3 (Comments on Novelty/Model System):

Overall this is good work which manages to find some novelty in an area of TB transcriptome analysis that has already been well explored by the authors and other groups.

Referee #3 (Remarks):

Maertzdorf et al. present the latest chapter in an impressive body of work performed to investigate host transcriptional responses to infection with *Mycobacterium tuberculosis* with the goal of identifying signature genes that could serve as the basis of potential biomarkers for diagnostic purposes. Similar studies have been conducted by other investigators, notably the O'Garra group, and generated substantial datasets that partially overlap or are similar to those discussed in this manuscript as the authors found in their meta-analysis of such datasets. The authors used two statistical tools to triage and rank the transcriptomics data: random forest and conditional inference trees. In so doing they identified four genes/transcripts that could reliably distinguish between cohorts of healthy individuals and tuberculosis patients. These genes are GBP1, IFITM3, P2RY14 and ID3. RT-PCR was used to test the robustness of the composite signature and overall the situation looks highly promising for the development of a point of care diagnostic tool.

Major comments

1. To date the authors have derived their tool from two well defined populations: healthy subjects and active tuberculosis cases. How does the 4-gene signature behave in cases of latent tuberculosis infection (LTBI)? Was the LTBI status known for the Indian cohort?
2. Along the same lines, it would be very interesting to know how the 4-gene signature behaves when tested blindly against a random collection of bloods including suspected TB cases and controls rather than simply against pre-identified samples.

Minor comments

The paper might benefit from moving supplemental figures 2 and 3 to the main text.
The writing is loose in places and also has formatting issues:

p.4 l.3 - should be "resource poor"

p.6 middle - the average reader will be unfamiliar with "Gini impurity" so some explanation is required.

p.9 l.1 - 2x "healthy"

p.10 l.5 - "classify TB from healthy" is neither precise nor elegant

p.12 - middle "important integrate part of our results" makes no sense so should be reworded.

p.12 - bottom "geographical" is misspelt.

p.15 - Accession number should be provided.

p.14 - Section on GeneXpert should reference one of the original source papers by the developers of the technology: e.g.

Am J Respir Crit Care Med. 2011 Nov 1;184(9):1076-84. doi: 10.1164/rccm.201103-0536OC.

Lancet. 2011 Apr 30;377(9776):1495-505. doi: 10.1016/S0140-6736(11)60438-8. Epub 2011 Apr 18.

J Clin Microbiol. 2010 Jul;48(7):2495-501. doi: 10.1128/JCM.00128-10. Epub 2010 May 26.

Paper explained section is missing.

References do not require numbering.

Labeling on axes of Figure 2C is illegible so a better version is required.

1st Revision - authors' response

19 October 2015

Responses to Referees

General responses:

The authors first of all thank all the Referees for carefully reviewing our manuscript and considering the significance of our study.

Below we address the major points brought up by the Referees and point-by-point responses to their remarks.

1) All reviewers have commented on the infection status (uninfected or LTBI) of individuals within the healthy control group, which is routinely assessed in clinical TB studies. We regret that we did not include this information in our manuscript before. We have now performed additional analyses including separate predictions for LTBI and uninfected controls. These new results show that there is no difference in the classification performance of our signatures between groups. We have included these results in the main text and additional figures in the Expanded View section.

2) A second major comment from both referees 2 and 3 concerns the inclusion of other diseases or TB-suspected cases. We fully agree that from a clinical perspective such a

comparison between true TB cases and other suspected cases is very important. This was also one of the major considerations why we included in our external cohort validations (Berry, Bloom, Kaforou and Maertzdorf sarcoidosis) separate analysis on TB versus other diseases. Unfortunately, we do not have direct access to such samples from other diseases or suspected TB cases to include in our RT-PCR validations. We would strongly support it when other investigators that do, would verify our signatures on such samples.

However, we also would like to emphasize that our study was initially set up to classify TB patients from healthy, and not for differential diagnosis of suspected disease. In our opinion, a simple “one size fits all” signature that would be able to distinguish TB from healthy and at the same time differentially diagnose TB from other (pulmonary) diseases will be very unlikely. We would rather favor an easy two parallel signature setup that when; a) the signature classifying healthy from probable TB tests positive, then; b) a second differential diagnosis signature (yet to be defined) would distinguish TB from other diseases. Future work in our lab is interested in developing an algorithm, which addresses these issues.

We have added the latter argumentation to the discussion section to better clarify our point of view.

3) We slightly changed the assignment of the different sample and datasets. In the original manuscript we described the separation of the Indian dataset into a training and validation set. In the revised version we changed this into training and test set, and use the term validation set only for the external and independent datasets.

Response to Referee 1

Referee #1 (Remarks):

This manuscript by Maertzdorf and co-workers describes the selection and use of human gene expression profiles for distinguishing tuberculosis (TB) patients from healthy individuals or patients with other pulmonary diseases. Based on data from large transcription profiling studies, the authors selected 4 genes, GBP1, IFITM3, P2RY14 and ID3 and obtained sensitivity scores of 88% with a specificity of around 75% to differentiate TB cases from healthy individuals. The selection of genes was based on signatures obtained from TB patients in Africa. Their profiles were then used for a training set and a validation set from patients from another geographic region (India). The authors also tested a 15 gene set and found that their selected four gene set provided equal or better discrimination of TB patients relative to patients with other diseases.

Overall, the study convincingly shows that RT-PCR results obtained from RNA preparations from patient's blood can help to reliably identify undiagnosed TB cases, suggesting that the described method could be developed into a promising diagnostic tools that might be used at the point of care units in addition to other tools for TB-case identification.

There was only one point that was not clear to me, which I may have missed: It was not clear to me which phenotypic criteria were originally used for defining individuals as TB patients or healthy individuals? Were the TB patients culture positive? Were the individuals tested by IGRA before being classified into the different categories of the study groups.

The criteria used for defining individuals as TB patients or healthy are now more clearly described in the inclusion and exclusion criteria in the Study design under Materials and Methods. TB patients were all GeneXpert positive.

Response to Referee 2

Referee #2 (Remarks):

Maertzdorf and colleagues extend the previous work conducted by them and others in identifying RNA signatures for active TB from whole blood. In this study they use a cohort of 120 active and 80 healthy controls from St. Johns Hospital in Bangalore, India. After curation they had 133 active TB and 76 healthy control samples which they divided into a training set (60 TB / 40 healthy) and validation set (53 TB / 36 healthy). They used both Random Forest and Conditional Inference tools to identify human transcripts with high predictive power. This led them to a 4 gene set and a 15 gene set. They further tested the 4 gene and 15 gene sets on 5 publically available studies: Berry (417 patients), Bloom 236 patients), Kaforou (327 patients), Maertzdorf (46 patients), and Dawany (36 patients). The results show that the 4 gene set achieves ROC AUC values of 0.90-0.99 in HIV-negative patients with TB vs healthy patients. For HIV positive patients with TB vs healthy patients it was 0.72-0.81, and for either HIV-positive or negative patients with TB versus those with other diseases it was 0.63 - 0.84.

Comments:

1. The WHO has identified a strong need for a rapid POC triage test to identify patients at high risk for TB who should undergo further evaluation, so this work is of very high significance. However, in the field one would anticipate that patients presenting will be ill with a mixture of TB and other diseases. Hence the fact that the 4 gene predictor set was derived using TB patients versus healthy controls and the test's relatively poor performance in TB versus other diseases (AUC 0.63-0.84) is a major limitation of the study

We are fully aware that being able to quickly and easily distinguish TB from other (pulmonary) diseases is of major clinical importance. However, our study was not initially designed for differential diagnoses. Please refer to general response point 2 for further explanation.

2. The authors used the training subpopulation (60 TB / 40 healthy patients) to derive a 15 gene set. They then tested and further refined the 15 gene set with the whole cohort of both the training and the validation sets (133 TB / 76 healthy) in order to obtain the 4 gene set. They never tell us how the 15 gene test performs only on the validation set, and of course the validation set was partially utilized to identify the 4 gene set.

The 15 gene set was not refined to obtain the 4 gene set. These signatures were derived from two separate models; the 15-gene set was built using random forest, and the 4-gene set derived from a conditional inference tree model.

The performance of the 15-gene set, built on the training set samples, was tested on the validation set (in the revised version termed test set) and was mentioned in the text (AUC=0.98, on bottom half of page 7). However, the 15 gene set, validated further on in the manuscript, was the one built on the combined training and test sets.

3. Please provide more information about the 80 healthy controls from Bangalore. Were they TST or IGRA positive for latent TB? How were they recruited? Were they actually patients seeking medical care or were they local students / health care workers?

We have added additional information on the recruitment and latent TB tests for healthy controls in the Study design section under Materials and Methods.

4. Human genetics is likely to have a strong influence on human RNA-based diagnostics for TB. It would be helpful if the authors stated the ethnicity of the patients in the 5 publically available data sets used for validation in Table 1

We agree that differences in genetic background of enrolled individuals in individual studies is an important factor in gene expression -based diagnostics tools and have included information on the major ethnic groups in each dataset in the table legend.

5. Similarly, the degree of HIV immunosuppression is also likely to have a strong influence on human RNA-based diagnostics for TB. It would be helpful if the authors stated the CD4 counts of the HIV-positive patients in the 5 publically available data setes used for validation in Table 1.

We agree that the degree of immunosuppression will likely influence gene expression measurements. However, we consider detailed clinical information on individuals within other publicly available datasets not a part of our study. We therefore did not include such details in our table but rather refer to the original publications.

Response to Referee 3

Referee #3 (Comments on Novelty/Model System):

Overall this is good work which manages to find some novelty in an area of TB transcriptome analysis that has already been well explored by the authors and other groups.

Referee #3 (Remarks):

Maertzdorf et al. present the latest chapter in an impressive body of work performed to investigate host transcriptional responses to infection with Mycobacterium tuberculosis with the goal of identifying signature genes that could serve as the basis of potential biomarkers for diagnostic purposes. Similar studies have been conducted by other investigators, notably the O'Garra group, and generated substantial datasets that partially overlap or are similar to those discussed in this manuscript as the authors found in their meta-analysis of such datasets. The authors used two statistical tools to triage and rank the transcriptomics data: random forest and conditional inference trees. In so doing they identified four genes/transcripts that could reliably distinguish between cohorts of healthy individuals and tuberculosis patients. These genes are GBP1, IFITM3, P2RY14 and ID3. RT-PCR was used to test the robustness of the composite signature and overall the situation looks highly promising for the development of a point of care diagnostic tool.

Major comments

1. To date the authors have derived their tool from two well defined populations: healthy subjects and active tuberculosis cases. How does the 4-gene signature behave in cases of latent tuberculosis infection (LTBI)? Was the LTBI status known for the Indian cohort?

We have performed additional analyses and now show in the revised manuscript also the classification performance for LTBI and uninfected controls separately. LTBI status for individuals in both the Indian cohort and the external RT-PCR datasets are now also given in the text and supplementary information.

2. Along the same lines, it would be very interesting to know how the 4-gene signature behaves when tested blindly against a random collection of bloods including suspected TB cases and controls rather than simply against pre-identified samples.

Please refer to general response point 2 for a detailed explanation.

Minor comments

The paper might benefit from moving supplemental figures 2 and 3 to the main text. The writing is loose in places and also has formatting issues:

We have moved the figures to the main text.

p.4 l.3 - should be "resource poor"

Corrected

p.6 middle - the average reader will be unfamiliar with "Gini impurity" so some explanation is required.

We have included an additional sentence to explain what these values mean, specifically in the context of our study.

p.9 l.1 - 2x "healthy"

Corrected

p.10 l.5 - "classify TB from healthy" is neither precise nor elegant

Has been reworded

p.12 - middle "important integrate part of our results" makes no sense so should be reworded.

Has been changed into "key part of our results".

p.12 - bottom "geographical" is misspelt.

Corrected

p.15 - Accession number should be provided.

Data have been uploaded to the GEO database and the accession nr. assigned is given at the end of the manuscript. GEO submission will be released to public status after acceptance of the manuscript.

*p.14 - Section on GeneXpert should reference one of the original source papers by the developers of the technology: e.g.
Am J Respir Crit Care Med. 2011 Nov 1;184(9):1076-84. doi: 10.1164/rccm.201103-0536OC.
Lancet. 2011 Apr 30;377(9776):1495-505. doi: 10.1016/S0140-6736(11)60438-8. Epub 2011 Apr 18.
J Clin Microbiol. 2010 Jul;48(7):2495-501. doi: 10.1128/JCM.00128-10. Epub 2010 May 26.*

We have changed the reference to GeneXpert to Boehme et al. 2011 (second referee suggestion)

Paper explained section is missing.

Section has been completed

References do not require numbering.

Reference numberings have been removed

Labelling on axes of Figure 2C is illegible so a better version is required.

The revised manuscript now contains an updated figure with better illegible labels.

3rd Editorial Decision

11 November 2015

Thank you for the submission of your revised manuscript to EMBO Molecular Medicine. We have now received the enclosed report from the referee who was asked to re-assess it. As this reviewer is fully supportive and has no additional comment, I am pleased to inform you that we will be able to accept your manuscript pending editorial final amendments.

Please submit your revised manuscript within 1 week in order for the article to be published in the January issue.

I look forward to seeing a revised form of your manuscript as soon as possible.